# Mitigating Hallucinations in LVLMs via Summary-Guided Decoding

**Kyungmin Min**[1]    **Minbeom Kim**[1]    **Kang-il Lee**[2]
**Dongryeol Lee**[2]    **Kyomin Jung**[1,2*]
[1]IPAI, SNU    [2]ECE, SNU
{kyungmin97,minbeomkim,4bkang,drl123,kjung}@snu.ac.kr

## Abstract

Large Vision-Language Models (LVLMs) have demonstrated impressive performance on multimodal tasks. However, they struggle with object hallucinations due to over-reliance on learned textual patterns, ignoring the provided image. To address this issue, we first investigate language priors in LVLMs. We observe two key findings: (1) Even when predicting image-related part-of-speech (POS) tokens, models increasingly rely on linguistic priors as the token sequences grow, thereby amplifying hallucinations. (2) Methods that directly control LVLM's output distribution to mitigate language priors can lead to a degradation in text quality or exacerbate hallucinations. Based on these insights, we propose Summary-Guided Decoding (SGD). This method naturally encourages the model to focus more on the image information, with control over only the image-related POS tokens for preserving text quality. Through experiments, we demonstrate that SGD achieves state-of-the-art performance on object hallucination benchmarks. Furthermore, while existing methods show a trade-off between precision and recall, SGD proves to be Pareto optimal in this respect. Lastly, we show that while existing methods suffer from text quality degradation due to such trade-offs, SGD preserves text quality to the maximum extent possible. This paper not only focuses on preventing object hallucination but also presents analysis and solutions aimed at maintaining the original properties of LVLMs.

## 1   Introduction

Large Vision-Language Models (LVLMs) have shown remarkable advancements by integrating the reasoning capabilities of Large Language Models (LLMs) to interpret visual knowledge [24, 3, 18]. Despite their significant utility, they suffer from a critical drawback known as *object hallucination*. This occurs when models produce responses that contradict the visual input, relying too heavily on language priors (i.e., language patterns learned during training) instead of the actual visual information [23, 16, 8]. This over-reliance on language priors intensifies when the LLM's fine-grained explanations are needed, e.g., models generate longer sequences, as shown in Figure 1, or encounter unseen visual inputs  [7]. In this paper, we aim to 1) investigate the fundamental analysis of language priors in LVLMs, 2) address limitations of existing methods and insights into potential solutions, and 3) propose a novel method that effectively removes object hallucination while preserving the original intent of the LVLM's response as much as possible.

First, we analyze language priors based on the distributional distance between the next token probabilities of LVLMs and LLMs, both conditioned on the same text sequence. When analyzing this by POS (part-of-speech) type, we observed a reasonably large divergence for image-related POS tokens

---

*Corresponding authors.

38th Conference on Neural Information Processing Systems (NeurIPS 2024).

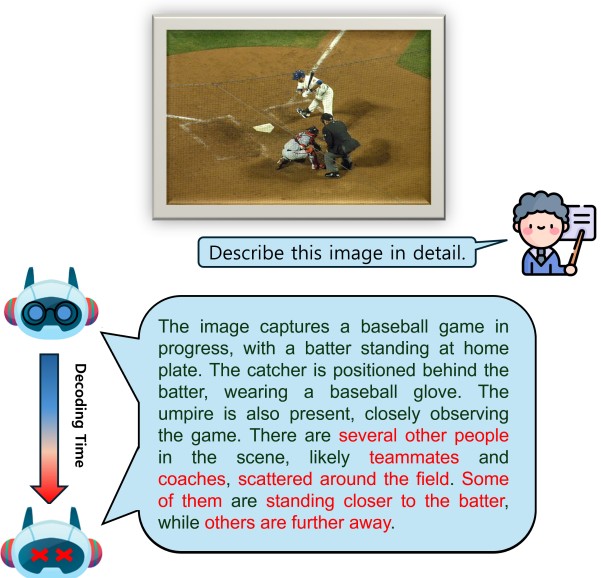

Figure 1: An example of LVLMs' hallucination. LVLMs hallucinate due to its over-reliance on previously generated text. The red fonts represent the hallucinatory content.

such as NOUN (e.g. boy, tree) and ADJ (e.g. green). Conversely, language-related POS tokens showed nearly identical distributions. This suggests that LVLMs reflect visual information within a linguistic template very similar to that of LLMs. Problematically, we discovered that even for these image-related POS tokens, the distributional distance rapidly decreases as the number of generated tokens increases. Consequently, the attention weight given to image tokens dramatically reduces, ultimately leading to frequent occurrences of object hallucination. We identify this phenomenon as an over-reliance on language priors.

A recently popular method to reduce this dependence on language priors is contrastive decoding. This approach emphasizes image-related tokens by subtracting the distribution of language prior-oriented models from the LVLM's output distribution. However, based on our analysis, two main issues are anticipated with this method: 1) The distribution of language-related POS tokens, necessary for maintaining linguistic quality, should be preserved. However, this information is also damaged, *leading to text quality degradation* (see Analysis 5.1). 2) As token length increases, the distributions of LVLM and LLM become increasingly similar for all tokens, gradually diminishing the effect of contrasting. In essence, using a language prior-biased distribution to guide a LVLM's original distribution towards a visual-oriented distribution results in numerous side effects. Therefore, we gain the insight that we should allow the LVLM to naturally reference visual information while limiting our influence to image-related POS tokens.

Building on this observation, we propose a simple yet effective method called **Summary-Guided Decoding (SGD)**. First, to preserve the text quality of LVLM, we intervene minimally in the decoding process. Specifically, we only refer to the *reference* when the next token is an image-related POS token. Here, the *reference* is a summarization of previously generated sentences, designed to reflect visual information while reducing the context length as much as possible. According to our analysis, with these summarized inputs, LVLM selects the next token while grounding in more visual information. Through this methodology, we can maintain the LVLM's language template almost intact while maximally reflecting image information for image-related POS tokens through summarized inputs.

Our experimental results demonstrate that SGD significantly surpasses all other decoding approaches in object hallucination benchmarks (e.g., up to +16.5% in $CHAIR_S$ and +19% in $CHAIR_I$). Furthermore, We add a recall axis to the precision-based evaluation method to assess the ability to produce accurate and fine-grained descriptions. The results reveal that contrastive decoding methods exhibit good accuracy but show lower recall than even greedy decoding, indicating 'repetition' and leading to significant degradation in text quality. In contrast, SGD demonstrated Pareto optimal performance in both precision and recall compared to existing methodologies, with this difference becoming

increasingly pronounced as token length increases. Additionally, we proved SGD's contribution by showing that POS control preserves LVLM's text quality almost entirely.

Our contributions are summarized as follows:

- We analyze the distributional distance by POS type to understand the decoding process of LVLMs. LVLM reflects visual information for image-related POS tokens on top of LLM's linguistic template. However, we also observed that as token length increases, the model tends to rely solely on language priors, even for image-related POS.

- Based on these findings, we propose a methodology called Summary-Guided Decoding (**SGD**). SGD refers to summarized token distributions only for image-related POS, aiming to reflect image information while preserving LVLM's text quality as much as possible.

- SGD demonstrates state-of-the-art performance in object hallucination benchmarks and Pareto optimal across all methods regarding the precision-recall trade-off. Additionally, unlike contrastive decoding, SGD is shown to preserve text quality almost entirely.

## 2 Language Priors in LVLMs

### 2.1 How to measure language priors in LVLMs

In LVLMs, language priors denote the model's over-dependence on learned textual patterns, where responses are generated based on these patterns without considering the provided image. From this perspective, we measure language priors based on the distributional distance between the next-token probabilities of LVLMs and LLMs, where LLM refers to the LVLM without a provided image. A larger distance indicates that the LVLM requires visual information to make predictions, suggesting a lower reliance on language priors. Conversely, a smaller distance suggests that the model is generating responses primarily based on textual patterns. To measure this distance between the probability distributions, we use Jensen-Shannon Divergence (JSD) [14]. A smaller JSD value implies a stronger influence of language priors, while a larger JSD indicates that the provided image is contributing more to the model's predictions.

### 2.2 Analysis of language priors by Part of Speech (POS) type

We generate 5,000 MSCOCO images [15] captions using LLAVA 1.5 7B model [18]. Specifically, we measure the JSD at each decoding step and average the JSD values for each Part-of-Speech (POS) types[2] up to 32 tokens. A key finding is the significant variation in divergence across different POS categories. As shown in Figure 2 (a), POS categories such as NOUN and ADJ, which rely more heavily on visual information, exhibit higher divergence. On the other hand, language-related POS types, like PART (e.g. particles such as "not, 's"), show much lower JSD. This indicates that LVLMs integrate visual information within a linguistic pattern highly aligned with LLMs. Another important observation, as shown in Figure 2 (b), is that even for image-related POS tokens, the distributional distance decreases significantly as token length increases, making the distribution of LVLMs and LLMs very similar. This suggests that even when image information is needed during decoding, models primarily rely on textual patterns. In other words, token length (or input length) has a significant impact on the language prior.

### 2.3 Longer token sequences amplify language priors in LVLMs

We observed that as token sequences grow longer, the model becomes increasingly dependent on language priors. To further investigate this effect, we analyze how token length influences LVLMs. We use MSCOCO 5,000 image captions for analysis of attention weight and object hallucinations (see Appendix A for details).

First, we measure the attention weights assigned to image tokens and text tokens at each decoding step. Figure 3 (a) shows that as the sequence length extends, the model progressively allocates less attention to image tokens, which encode essential visual information. The reduction in attention to image tokens causes the model to depend more generated text rather than visual inputs to predict the

---

[2]We utilized the Spacy model (en_core_web_sm) for POS tagging

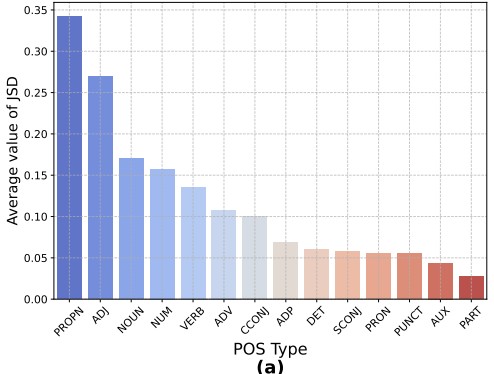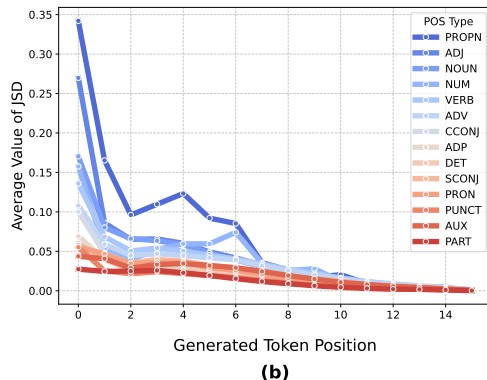

Figure 2: **(Left)** The average JSD between the LVLM and the LLM for each POS category up to 32 tokens. **(Right)** The average JSD between the LVLM and the LLM for each POS category across intervals, with each interval consisting of 32 tokens.

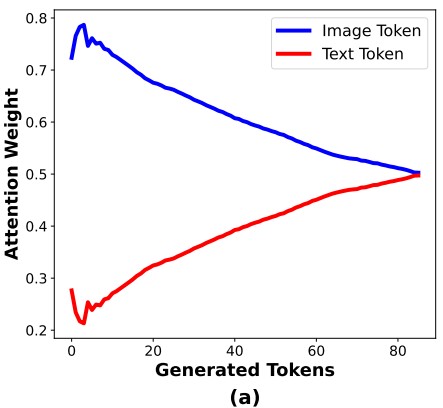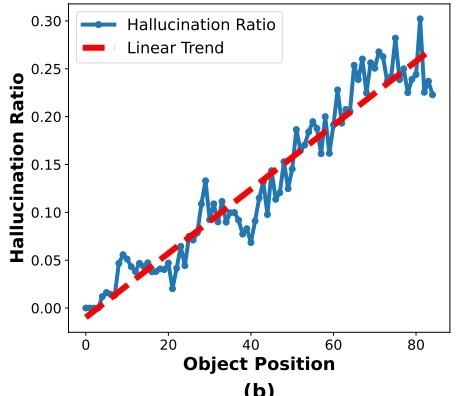

Figure 3: **(Left)** Attention weights of image tokens and text tokens at each decoding step (or token length). **(Right)** Object hallucination ratio at each generated token position.

next token. This finding aligns with our observation in Section 2.2, where longer sequences reinforce reliance on language priors.

To assess the role of input length in hallucination, we evaluate the occurrence object hallucination as a function of token length. Figure 3 (b) shows a clear positive correlation between input length and the likelihood of object hallucinations, indicating that longer text generation increases the chances of hallucination. We conclude that this phenomenon is driven by over-reliance on language priors, which amplifies hallucinations in LVLMs.

Recent research has utilized contrastive decoding to reduce the model's dependence on language priors for mitigating hallucinations [25, 21, 4, 21]. However, our detailed analysis of the trade-offs between contrastive decoding and token length (see Analysis 5.1) suggests that allowing the LVLM to naturally draw on more visual contents by token length control, while constraining intervention to image-related POS tokens, strikes an effective balance between factuality and text quality.

## 3 Summary-Guided Decoding

In Section 2, we identified that an increase in input length results in greater reliance on language priors, thereby exacerbating hallucinations in LVLMs. To address this, we present Summary-Guided Decoding (SGD), an efficient method for controlling the length of conditioning input during decoding. In this approach, after generating each sentence, the previous text is summarized to capture the critical information from earlier outputs. This approach effectively reduces the input length, allowing the model to stay more focused on the provided image when generating subsequent tokens while maintaining the crucial context.

Using summarized inputs can reduce contextual information, which may cause discrepancies with the language patterns previously learned by the model. This can result in distributional shifts that

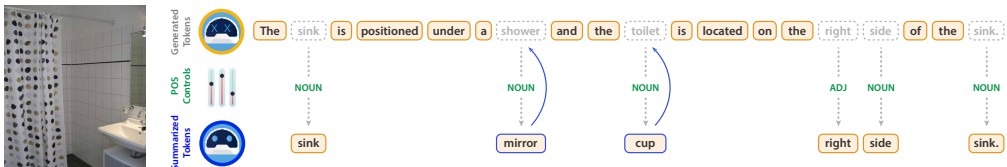

Figure 4: Illustration of our Summary-Guided Decoding.

weaken the model's language modeling capabilities, ultimately degrading the quality of the generated text. To address this, we preserve the original distribution for tokens related to language modeling while using summary-guided decoding to control only image-related POS tokens[3], ensuring factual accuracy and high text quality. Our method is illustrated in Figure 4, and a detailed explanation of our method is in Appendix C.

We introduce two variations of Summary-Guided Decoding for summary model usage. The first approach takes advantage of the instruction-following abilities inherent in LVLMs, which are based on LLMs. By providing summary instructions directly to the LVLM, this method allows the model to perform summary-guided decoding without incurring additional memory costs. However, a limitation of this approach is the increased inference time, as the LVLM generates its summaries during the process. This also limits the ability to support parallel decoding, resulting in slower performance. To address these challenges, we distill the summarization capability into a smaller, more efficient model, Flan-T5-base [2] (please refer to Appendix B). This model significantly reduces computational overhead while preserving the advantages of input length control. We report results for both the **SGD with the Self-Summarization (SGD-S)** and the **SGD with the Distilled-T5 model (SGD-D)**, demonstrating the trade-offs between efficiency and performance.

## 4 Experiment

### 4.1 Experiment settings

**Datasets and Evaluation Metrics.** We employ the Caption Hallucination Assessment with Image Relevance (CHAIR) [20] for evaluating object hallucination. We generate descriptions for 200 images from the MSCOCO 2014 validation dataset [15] prompted with ''`Please describe this image in detail.`''. CHAIR consists of two variants: $\text{CHAIR}_I$, which calculates the percentage of hallucinated objects out of all objects mentioned in the caption, and $\text{CHAIR}_S$, which measures the percentage of captions that contain at least one hallucinated object. Additionally, to complement the precision-based CHAIR metric, we include a Recall metric for the evaluation.

$$\text{CHAIR}_I = \frac{|\{\text{hallucinated objects}\}|}{|\{\text{all mentioned objects}\}|}, \quad \text{CHAIR}_S = \frac{|\{\text{captions with hallucinated objects}\}|}{|\{\text{all captions}\}|}, \quad \text{Recall} = \frac{|\{\text{correct mentioned objects}\}|}{|\{\text{ground truth objects}\}|}.$$

Additionally, we employ the Sentence-level Hallucination Ratio (SHR), a GPT-4-based evaluation metric, for a more holistic assessment of hallucinations. This metric captures object existence hallucinations and those related to object relations, attributes, and movements [22]. We generate descriptions for 200 images from the VG dataset [9], using the same prompts as in the CHAIR metric. Specifically, SHR leverages GPT-4[4] to compare the model's responses with the manually annotated descriptions from the VG dataset, evaluating each response on a sentence-by-sentence to identify potential hallucinations accurately.

**Baseline LVLMs.** In LVLMs, two prominent methods for aligning text and vision modalities are the projection layer-based approach and the learnable query-based approach [12, 24, 1, 17]. In our experiments, we utilized representative models for each method: LLAVA-1.5 7B [18] and InstructBLIP 7B [3].

**Baseline Methods.** We include various decoding methods as baseline approaches in our study. We use greedy decoding, nucleus sampling, and beam search for traditional methods. In addition, we incorpo-

---

[3]As shown in figure 2, we select PROPN, ADJ, NOUN and NUM as image-related POS.

[4]We used GPT-4o model for hallucination judgement.

Table 1: Results on CHAIR evaluation. The best performances within each setting are bolded. Max new tokens are 512.

| Method | LLAVA-1.5 | | | InstructBLIP | | |
|---|---|---|---|---|---|---|
| | CHAIR$_S$ ↓ | CHAIR$_I$ ↓ | Recall ↑ | CHAIR$_S$ ↓ | CHAIR$_I$ ↓ | Recall ↑ |
| Greedy | 51.5 | 13.7 | 79.1 | 60.5 | 25.3 | 68.1 |
| Nucleus | 53.0 | 14.4 | 76.9 | 57.5 | 24.6 | 65.0 |
| Beam | 47.5 | 12.5 | **79.2** | 54.0 | 16.3 | **74.1** |
| OPERA | 46.0 | 13.4 | 78.3 | 50.0 | 14.0 | **74.1** |
| VCD | 58.0 | 16.4 | 77.8 | 54.0 | 17.8 | 71.6 |
| ICD | 45.5 | 13.4 | 77.2 | 62.5 | 20.0 | 71.2 |
| M3ID | 44.5 | 12.0 | 76.1 | 66.0 | 27.2 | 66.9 |
| **SGD-D (Ours)** | **42.5** | 11.8 | 77.8 | 43.5 | 14.4 | 68.3 |
| **SGD-S (Ours)** | 43.0 | **11.1** | 79.1 | **43.0** | **13.6** | 69.5 |

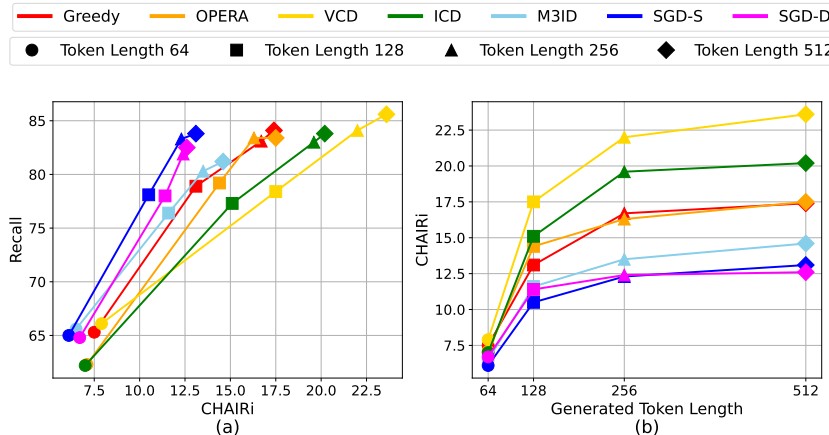

Figure 5: **(Left)** A position closer to the top-left indicates an optimal balance between factuality and recall. **(Right)** Trade-off between generated token length and hallucination (lower is better).

rate contrastive decoding techniques including Visual Contrastive Decoding (VCD) [11], Instruction Contrastive Decoding (ICD) [21], and Multi-Modal Mutual Information Decoding (M3ID) [4], which is designed to address hallucinations. Lastly, we include OPERA [6], a beam search-based method designed to counteract the model's tendency to focus heavily on specific anchor tokens.

## 4.2  Main Results

**Results on CHAIR metric** As shown in Table 1, SGD significantly improves overall baseline methods in the CHAIR$_S$ and CHAIR$_I$ for the LLAVA 1.5 and InstructBLIP. Specifically, compared to Greedy decoding, SGD-S achieves a 16.5% improvement in CHAIR$_S$ and a 19% improvement in CHAIR$_I$ on LLAVA 1.5. On InstructBLIP, the improvements are even more pronounced, with a 28.9% improvement in CHAIR$_S$ and a 46.2% improvement in CHAIR$_I$. Notably, the Recall remains unchanged for LLAVA 1.5. It even improves for InstructBLIP, which indicates that CHAIR metrics are not enhanced due to mentioning fewer objects but rather more accurate predictions.

CHAIR is a precision-based metric, which means it can be hacked by generating shorter captions or fewer objects. To enable a fair evaluation of object hallucination across different methods, we fix the generated token lengths at 64, 128, 256, and 512 from short to long text generation in LLAVA 1.5. As illustrated in Figure 5 (a), SGD-S attains the most favorable balance between factual accuracy and recall, irrespective of whether the descriptions are short or long. Furthermore, figure 5 (b) shows that SGD-S exhibits a lower degree of object hallucination across all fixed token lengths. These findings suggest that our proposed SGD method offers a robust and generalizable approach for ensuring factual decoding, applicable to both short and long descriptions.

**Results on Sentence-level Hallucination metric** Table 2 demonstrates that SGD-S shows the lowest hallucination sentence ratio in LLAVA 1.5, while in InstructBLIP, SGD-D achieves

Table 2: Results on Sentence-Hallucination Ratio (SHR), Sentence per Image (SPI),and n-gram repetition. The best performances within each setting are bolded. Max new tokens are 512.

| Method | LLAVA-1.5 | | | | InstructBLIP | | | |
|--------|---------|-----|--------|--------|---------|-----|--------|--------|
| | SHR ↓ | SPI | 1-gram ↑ | 2-gram ↑ | SHR ↓ | SPI | 1-gram ↑ | 2-gram ↑ |
| Greedy | 43.3 | 5.00 | 62.9 | 93.2 | 66.9 | 3.31 | 97.2 | 99.9 |
| OPERA | 42.0 | 4.74 | 63.8 | 92.4 | **51.7** | 4.96 | 64.1 | 91.6 |
| VCD | 52.0 | 5.18 | 67.0 | 95.6 | 60.0 | 4.56 | 79.5 | 97.4 |
| ICD | 50.2 | 4.93 | 65.5 | 94.3 | 61.1 | 5.41 | 76.6 | 96.3 |
| M3ID | 46.4 | 5.02 | 66.4 | 94.9 | 71.7 | 3.65 | 96.9 | 99.9 |
| SGD-D | 41.7 | 5.08 | 61.4 | 91.7 | 58.9 | 3.97 | 83.7 | 98.3 |
| SGD-S | **40.8** | 5.03 | 61.4 | 92.0 | 60.5 | 4.01 | 83.9 | 98.3 |

the second lowest hallucination ratio, just after OPERA. However, OPERA relies on beam search, which is computationally more expensive than our approach. Additionally, the n-gram repetition results indicate that our method avoids repeating specific words in its outputs.

## 5 Analysis

### 5.1 Analysis of SGD and Contrastive Decoding

In this section, we provide an in-depth analysis of SGD and contrastive decoding in LLAVA 1.5. Two key research questions guide the analysis. **RQ1**: *Is significantly deviating from language priors always beneficial?* **RQ2**: *When language priors have heavily influenced or distorted the original distribution, can contrastive decoding still guide the model to produce factually accurate outputs?*

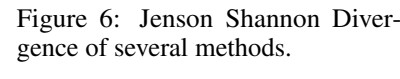

Figure 6: Jenson Shannon Divergence of several methods.

To investigate the relationship between each method and language priors, we compute JSD at each decoding step, following the approach described in Section 2.1. We generate descriptions for 200 images from the MSCOCO 2014 validation dataset. Additionally, we use the CHAIR metric to assess factuality and the GPT-4 model to evaluate text quality. Text quality is rated on a scale of 1 to 5 (more details in Appendix D)

As shown in Figure 6, the JSD for 256 tokens reveals that M3ID significantly reduces the influence of language priors. However, as presented in Table 3, text quality drops considerably from 4.85 to 2.39 when generating up to 64 tokens compared to 256 tokens, a decline of about 50.7%. This suggests that aggressively reducing language priors can lead to a notable decrease in language modeling performance. Interestingly, with greedy decoding, the JSD at 256 tokens is very low, indicating that the model heavily relies on the generated text, overlooking visual content. When comparing ICD with CHAIR, we observe that ICD increases hallucinations more

Table 3: CHAIR metric and Text Quality in various generated token lengths. Denote CHAIR$_S$ as $C_S$, CHAIR$_I$ as $C_I$ and Text Quality as TQ.

| Method | Token length 64 | | | Token length 256 | | |
|--------|---------|---------|--------|---------|---------|--------|
| | Cs ↓ | Ci ↓ | TQ ↑ | Cs ↓ | Ci ↓ | TQ ↑ |
| Greedy | 27 | 7.5 | 4.97 | 67.5 | **16.7** | 4.46 |
| ICD | 21.5 | 7 | 4.92 | 71 | **19.6** | 4.67 |
| M3ID | 20.5 | 6.5 | **4.85** | 62 | 13.5 | **2.39** |
| SGD | 22.5 | 6.1 | 4.93 | 54 | **12.3** | 3.75 |

than greedy decoding. This aligns with previous findings that contrastive decoding encourages more diverse responses [13]. As a result, this diversity can generate less related text to the image, thus amplifying hallucinations. In contrast, our method selectively moves away from the language prior for image-related POS tokens, while allowing the prior to influence other tokens. This approach results in a more natural output distribution. Additionally, our approach significantly reduces object hallucination and maintains better text quality than M3ID, which eliminates language priors more aggressively as token length increases.

Table 4: Ablation study in terms of summary quality and POS Control

| | CHAIR$_S$ ↓ | CHAIR$_I$ ↓ | Recall ↑ | Text Quality ↑ | 1-gram ↓ | 2-gram ↓ |
|---|---|---|---|---|---|---|
| Greedy Decoding | 51.5 | 13.7 | 79.1 | 4.9 | 46.72 | 10.99 |
| *Summary Models* | | | | | | |
| Distilled-Flan-T5-base(248M) | 42.5 | 11.8 | 77.8 | 4.8 | 49.86 | 14.53 |
| LLAVA 1.5(7B) | 43 | 11.1 | 79.1 | 4.81 | 49.15 | 13.28 |
| GPT-4o [19] | 43 | 10.3 | 78 | 4.77 | 51.9 | 16.41 |
| *POS Control in SGD* | | | | | | |
| ALL POS | 39 | 10.1 | 75.8 | 4.06 | 64.85 | 33.41 |
| Image-related POS | 43 | 11.1 | 79.1 | 4.81 | 49.15 | 13.28 |

## 5.2 Ablation study

In this section, We use LLAVA 1.5 to generate descriptions for 200 images from the MSCOCO 2014 validation dataset, same evaluation metric as in section 5.1

**Summary Models.** We conduct an ablation experiment based on the quality of summaries within the context of SGD. We generate summaries using three different models, each with increasing computational cost. Through Table 4, we observed that the effect of summarization quality is similar between the three models. This suggests that both SGD-D and SGD-S demonstrate satisfactory summarization quality.

**POS Control.** We analyze the impact of SGD on object hallucination and text quality when applied to all tokens compared to targeting only image-related POS tokens. As detailed in Table 4, applying SGD in both scenarios demonstrated a reduction in object hallucination compared to original decoding. However, when SGD was applied to all tokens, we observed a decline in text quality, with notable increases in repetition and degradation in object recall compared to original decoding. This observation indicates that applying SGD to all POS tokens weakens the model's language modeling capabilities, diminishing the ability to produce detailed descriptions. In contrast, when SGD is selectively applied to image-related POS tokens, the text quality and the repetition of the text remained comparable to the original decoding.

## 6    Related works

**Mitigating Language Priors in LVLMs.** Large Vision-Language models (LVLMs) extend pre-trained Large Language Models (LLMs) by incorporating visual tokens, enabling them to process visual content [17, 3, 24]. In LVLM architectures, the language model is significantly larger than the vision model, creating an imbalanced structure where the language model exerts more significant influence. As a result of this imbalance, the model tends to rely on linguistic patterns rather than adequately considering the visual information provided, a phenomenon known as the language prior problem [5, 7, 10]. To address this issue, several studies have explored contrastive decoding techniques to mitigate the model's over-reliance on language priors. Visual Contrastive Decoding (VCD) [11] works by utilizing distorted images, which amplify the language prior, while Instruction Contrastive Decoding (ICD) [21] introduces misleading instructions to achieve a similar effect. Both methods aim to reduce the language prior's dominance by leveraging these amplified conditions to adjust the model's behavior. Additionally, Multi-Modal Mutual Information Decoding (M3ID) [4] identified that as the token length increases, the model dilutes visual information, leading to a more substantial reliance on language priors. To counter this, M3ID applies more assertive contrastive decoding techniques as the token length grows to calibrate the model's over-reliance on language priors.

## 7    Conclusion

This paper proposes a simple yet effective Summary-Guided Decoding method, based on the fundamental analysis of language priors in LVLMs using part-of-speech tags. Our method summarizes previous tokens to reduce token length, thereby naturally guiding the model to rely more on the image. Additionally, by controlling only the image-related POS tokens, we prevent degradation in text quality. Experimental results show that our method significantly reduces object hallucination and achieves the most optimal balance between factual accuracy and recall in both short and long description tasks.

## Acknowledgements

This research was supported by the MSIT(Ministry of Science and ICT), Korea, under the ITRC(Information Technology Research Center) support program(IITP-2024-RS-2024-00437633) supervised by the IITP(Institute for Information Communications Technology Planning Evaluation). This work was partly supported by Institute of Information & communications Technology Planning & Evaluation (IITP) grant funded by the Korea government(MSIT) [RS-2021-II211343, Artificial Intelligence Graduate School Program (Seoul National University) & RS-2021-II212068, Artificial Intelligence Innovation Hub (Artificial Intelligence Institute, Seoul National University)], and the BK21 FOUR program of the Education and Research Program for Future ICT Pioneers, Seoul National University in 2024. K. Jung is with ASRI, Seoul National University, Korea. The Institute of Engineering Research at Seoul National University provided research facilities for this work.

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

# A    Experiment settings

For the token-level assessment in object hallucination, we generated descriptions for 5000 images from the MSCOCO dataset [15] and annotated each token to determine whether it represented an object hallucination, defined as an object not present in the image, using the CHAIR metric pipeline [20] for evaluation.

# B    Distill Flan-T5-base model

First, we employed LLAVA 1.5 to perform Summary-Guided Decoding with Self-Summarization when generating descriptions for 5,000 images from the MSCOCO dataset. During this process, LLAVA 1.5 iteratively summarized the previous sentence, and we saved each previous sentence along with its corresponding summarized sentence as a pair. This paired dataset was then used to fine-tune the Flan-T5-base model with the prompt "What is a summary of this text?" for training purposes.

# C    Detail explanation of Summary-Guided Decoding

In LVLMs, next-token predictions often result in sub_tokens (partial tokens), making accurate Part-of-Speech (POS) tagging more challenging. To mitigate this issue, greedy decoding is used to form complete tokens (words) by predicting future tokens until a full sentence is generated. This approach enables accurate POS tagging for the current sub_token. While this method significantly improves POS tagging accuracy, it comes with an increased decoding cost, as forming a complete sentence requires additional token predictions. Nevertheless, the accuracy improvements make this trade-off worthwhile. After POS tagging the current sub_token, if the POS tag belongs to an image-related, the input is considered a summarized version of the previously generated sentence. However, if the POS is associated with language modeling, the original input is retained.

# D    GPT-4o Prompt for text quality evaluation

"' Task Description: You will be given one caption written for a given image. Your task is to rate the caption on one metric. Please make sure you read and understand these instructions carefully. Please keep this document open while reviewing, and refer to it as needed. The output format should look as follows: Score: [RESULT] (an integer number between 1 and 5). Please do not generate any other opening, closing, and explanations.

Evaluation Criteria: Text Quality (1-5) - Evaluate how well-written the caption is. A high-quality caption is clear, concise, grammatically correct, and well-structured.

Evaluation Steps: 1. Read the caption carefully and evaluate its clarity, grammar, and overall readability. 2. Check for any awkward phrasing, grammatical errors, or unnecessary complexity. 3. Assign a score for text quality on a scale of 1 to 5, where 1 is the lowest and 5 is the highest based on the Evaluation Criteria.

Given Caption: Caption

Score: '"

# E    Limitation

In this paper, we evaluated our method solely on the captioning task due to the lack of multimodal tasks that require continuous image-guided generation. As more generation tasks become available, we will be able to conduct a more detailed analysis of our approach. Additionally, we reported results only on LLAVA 1.5 and InstructBLIP. For future work, we plan to extend our evaluation to a broader range of models.

# F    CHAIR metric on various token length

In this section, we report CHAIR metric based on various generated token length.

Table 5: Results on CHAIRs, CHAIRi, and Recall

| Generated Token Length | Method | CHAIRs | CHAIRi | Recall |
|---|---|---|---|---|
| 64 | Greedy | 27 | 7.5 | 65.3 |
| 64 | Nucleus | 31.5 | 9.8 | 58.9 |
| 64 | Beam | 20 | 5.9 | 62.5 |
| 64 | VCD | 24.0 | 7.9 | 66.1 |
| 64 | ICD | 21.5 | 7.0 | 62.2 |
| 64 | M3ID | 20.5 | 6.5 | 65.6 |
| 64 | Opera | 22.5 | 7.1 | 62.3 |
| 64 | SGD-S | 22.5 | 6.1 | 65.0 |
| 64 | SGD-D | 24 | 6.7 | 64.8 |
| 128 | Greedy | 53 | 13.1 | 78.9 |
| 128 | Nucleus | 56.5 | 16.5 | 74.2 |
| 128 | Beam | 50.5 | 13.3 | 78.3 |
| 128 | VCD | 63.0 | 17.5 | 78.4 |
| 128 | ICD | 56.0 | 15.1 | 77.3 |
| 128 | M3ID | 46.5 | 11.6 | 76.4 |
| 128 | Opera | 49.5 | 14.4 | 79.2 |
| 128 | SGD-S | 43.5 | 10.5 | 78.1 |
| 128 | SGD-D | 43.5 | 11.4 | 78.0 |
| 256 | Greedy | 67.5 | 16.7 | 83.1 |
| 256 | Nucleus | 78 | 20.9 | 82.8 |
| 256 | Beam | 70 | 16.2 | 81.6 |
| 256 | VCD | 82.5 | 22.0 | 84.1 |
| 256 | ICD | 71 | 19.6 | 83.0 |
| 256 | M3ID | 62 | 13.5 | 80.3 |
| 256 | Opera | 64.5 | 16.3 | 83.4 |
| 256 | SGD-S | 54 | 12.3 | 83.3 |
| 256 | SGD-D | 56.5 | 12.4 | 81.9 |
| 512 | Greedy | 69.5 | 17.4 | 84.1 |
| 512 | Nucleus | 80 | 22.0 | 83.8 |
| 512 | Beam | 71.5 | 17.4 | 82.3 |
| 512 | VCD | 83.0 | 23.6 | 85.6 |
| 512 | ICD | 73.0 | 20.2 | 83.8 |
| 512 | M3ID | 65.5 | 14.6 | 81.2 |
| 512 | Opera | 66.5 | 17.5 | 83.4 |
| 512 | SGD-S | 59 | 13.1 | 83.8 |
| 512 | SGD-D | 61.5 | 12.6 | 82.5 |

# G    About Reproduction

We used the code provided by the authors for VCD and OPERA, while M3ID and ICD were implemented from scratch due to the lack of public code. For VCD, OPERA, and ICD, we used the hyperparameters as specified in their respective papers. Since only LLAVA 1.5's hyperparameters were reported for M3ID, we applied these hyperparameters to both LLAVA 1.5 and InstructBLIP for our experiments. All the experiments are conducted using 1 NVIDIA RTX A6000 GPU.

# H    Societal impact

Object hallucination in multimodal systems poses a significant challenge on the path toward AGI, particularly in safety-critical applications such as autonomous driving. Reducing object hallucination is expected to be a crucial step in ensuring the development of safer and more reliable AI systems.

