# OpenReview forum: "Mitigating Hallucinations in LVLMs via Summary-Guided Decoding"
_NeurIPS.cc/2024/Workshop/SafeGenAi — SafeGenAi Poster_

### Official Review · Reviewer_AwB2 · 2024-10-09
**This paper proposes Summary-Guided Decoding (SGD), a method to mitigate object hallucinations in Large Vision-Language Models (LVLMs) by focusing on image-related POS tokens, achieving state-of-the-art performance without degrading text quality.**

**Rating:** 8
**Confidence:** 4

**Review:**

Pros :
- The paper offers a solid analysis of the limitations of LVLMs and presents a well-researched solution through the Summary-Guided Decoding (SGD) method.
- The SGD method is innovative, targeting image-related POS tokens for control and reducing language priors without sacrificing text quality.
- The experiments demonstrate a significant reduction in hallucination rates while maintaining or improving text quality, achieving Pareto optimal performance on both precision and recall.

Cons:
-  The paper does not extend its evaluation beyond two LVLMs (LLAVA and InstructBLIP), leaving out other popular models in the field, which could provide broader insights into its applicability
- Although SGD is presented as computationally efficient, there is little quantitative analysis on its inference time relative to other models, especially for longer sequences.

---

### Official Review · Reviewer_Nmzh · 2024-10-09

**Rating:** 6
**Confidence:** 4

**Review:**

- JSD between which and which?
- "POS categories such as NOUN and ADJ, which rely more heavily on visual information, exhibit higher divergence." I think this is just because NOUN and ADJ are all very different in the embedding space. And, in fact, there are a large number of NOUN and ADJ in the list. We need to fix the measure. JSD is probably not enough.

---

### Official Review · Reviewer_p4gr · 2024-10-10
**Addressing Hallucinations in LVLMs Through Context Shortening with Summary Models**

**Rating:** 7
**Confidence:** 5

**Review:**

**Summary**:
This paper explores how LVLM tend to forget image information and increasingly rely on language priors over time, leading to amplified hallucinations. To mitigate this issue, it propose using a summary model to shorten the context when decoding image-related POS tokens. Their approach achieves superior performance on the CHAIR metric.

**Strengthens**:
The paper offers valuable insights into how the forgetting of image information amplifies hallucinations during generation and proposes summary-guided decoding which effectively mitigates object hallucinations.

**Weaknesses**:
1. The study would benefit from incorporating additional object hallucination metrics, such as POPE, as well as qualitative results to support its claims.
2. Utilizing a summary model for POS token generation may lead to redundant outputs and omit important details.
3. Further analysis on latency and computational cost is needed to address concerns about the method's practical efficiency.